# Illegitimate Recombination between Duplicated Genes Generated from Recursive Polyploidizations Accelerated the Divergence of the Genus *Arachis*

**DOI:** 10.3390/genes12121944

**Published:** 2021-12-01

**Authors:** Shaoqi Shen, Yuxian Li, Jianyu Wang, Chendan Wei, Zhenyi Wang, Weina Ge, Min Yuan, Lan Zhang, Li Wang, Sangrong Sun, Jia Teng, Qimeng Xiao, Shoutong Bao, Yishan Feng, Yan Zhang, Jiaqi Wang, Yanan Hao, Tianyu Lei, Jinpeng Wang

**Affiliations:** 1Center for Genomics and Computational Biology, School of Life Sciences, North China University of Science and Technology, Tangshan 063000, China; 15131370810@163.com (S.S.); yuxianli120@gmail.com (Y.L.); wangjy_pg@163.com (J.W.); wechendan@163.com (C.W.); zhenyiwang0301@163.com (Z.W.); gwn-06@163.com (W.G.); yuanmin308@163.com (M.Y.); zhanglan1374@sohu.com (L.Z.); wlsh219@126.com (L.W.); sang123sang_3g@sina.cn (S.S.); tengjiaxinlang@sina.com (J.T.); xiaoqm2020@163.com (Q.X.); bst15530357750@163.com (S.B.); fengyishan123@163.com (Y.F.); zhangyan1220zj@163.com (Y.Z.); wangjiaqiky@163.com (J.W.); hyn712712@163.com (Y.H.); 2University of Chinese Academy of Sciences, Beijing 100049, China; 3State Key Laboratory of Systematic and Evolutionary Botany, Institute of Botany, Chinese Academy of Sciences, Beijing 100093, China

**Keywords:** *Arachis*, polyploidization, duplicated genes, gene conversion, subgenome

## Abstract

The peanut (*Arachis hypogaea* L.) is the leading oil and food crop among the legume family. Extensive duplicate gene pairs generated from recursive polyploidizations with high sequence similarity could result from gene conversion, caused by illegitimate DNA recombination. Here, through synteny-based comparisons of two diploid and three tetraploid peanut genomes, we identified the duplicated genes generated from legume common tetraploidy (LCT) and peanut recent allo-tetraploidy (PRT) within genomes. In each peanut genome (or subgenomes), we inferred that 6.8–13.1% of LCT-related and 11.3–16.5% of PRT-related duplicates were affected by gene conversion, in which the LCT-related duplicates were the most affected by partial gene conversion, whereas the PRT-related duplicates were the most affected by whole gene conversion. Notably, we observed the conversion between duplicates as the long-lasting contribution of polyploidizations accelerated the divergence of different *Arachis* genomes. Moreover, we found that the converted duplicates are unevenly distributed across the chromosomes and are more often near the ends of the chromosomes in each genome. We also confirmed that well-preserved homoeologous chromosome regions may facilitate duplicates’ conversion. In addition, we found that these biological functions contain a higher number of preferentially converted genes, such as catalytic activity-related genes. We identified specific domains that are involved in converted genes, implying that conversions are associated with important traits of peanut growth and development.

## 1. Introduction

The peanut (*Arachis hypogaea* L.), known as the “longevity fruit”, is the leading oil and food crop among the legume family and is in seed oil (~46–58%) and protein (~22–32%) [1,2]. Peanut products are rich in fat and protein, which are essential for eradicating malnutrition and ensuring food security, which directly reflects the value-added effect of comprehensive processing and utilization [2,3,4]. The worldwide area dedicated to peanut cultivation covers about 23 million hectares and the vast majority of peanuts (>95%) are grown in Asia and Africa, with an annual production of nearly 42 million tons [5]. The peanut originated from South America and belongs to the genus *Arachis* [6], which contains 81 species and can be divided into nine sections according to the morphological characteristics, geographical distribution and hybrid affinity [2,7]. In the section *Arachis*, there are mostly wild diploid species (2*n* = 2*x* = 20), with only two tetraploids (AABB, 2*n* = 4*x* = 40), namely, the wild (*Arachis monticola*) and cultivated (*A. hypogae**a*) species [8].

Polyploidization is a cataclysmic genomic event, which can create a large number of duplicate gene pairs which are separated and located in homoeologous chromosomes and provide basic materials for the genetic innovation of species and even triggers speciation and diversification processes [9,10,11,12]. Immediately after polyploidization, dramatic genome reshuffling and duplicated gene losses may often occur [13,14,15,16]. Through nucleotide mutation, the long-lasting retained duplicate genes provide functional innovation over millions of years and eventually lead to novel functions (neofunctionalization), or the subdivision of ancestral functions (subfunctionalization), or mixture of both (subneofunctionalization) [17,18,19]. In addition to mutation, duplicated genes can interact with each other directly through DNA recombination to achieve genetic innovation [20,21,22]. After the core eudicot common hexaploidization event (ECH), which can be dated to ~115–130 million years ago (MYA) [23,24], all *Arachis* genomes shared a tetraploidization event (LCT) with other legumes of soybean (*Glycine max*) and barrel medic (*Medicago truncatula*) about 60 MYA [2,3,4,5,6,25,26,27,28], producing thousands of duplicated genes in the extant *Arachis* genomes [26]. Although Zhuang et al. questioned the identity of one of the diploid ancestors (*Arachis duranensis*, AA genome) in tetraploid peanuts [2,29], it is undeniable that the hybridization between two diploids, the AA genome (such as *A. duranensis* or a close relative) and the BB genome (*Arachis ipaensis*), formed the wild allotetraploid *A. monticola* and, after further domestication, the cultivated tetraploid *A. hypogae**a* [6,7,30,31,32,33,34,35]. A large number of duplicated genes generated from the hybridization (or recent allo-tetraploidization, PRT) of diploid *Arachis* genomes are also exhibited in different subgenomes of tetraploid peanut genomes [2,4,5,28]. The recursive polyploidization events of *Arachis* provide an innovative material for peanut evolution and diversity formation [4,27].

Genetic recombination plays an important role in DNA repair and crossovers between homologous chromosomes (or DNA segments) are a major driving force of biological evolution [9,36]. The recombination between homologous chromosomes is often called homologous recombination, while the recombination between homoeologous chromosomes (generated from polyploidization) is considered an abnormal recombination, which is called “illegitimate recombination” [37]. DNA genetic information can be reciprocally or symmetrically exchanged between homologous sequences during the meiotic and mitotic recombination of plants [38]. Gene conversion results from nonreciprocal recombination, which involves the unidirectional transfer of one gene (or DNA segment) locus to its paralogous counterparts [39]. Gene conversion between duplicated genes (or homoeologous chromosomes) generated from whole-genome duplication (WGD) has been discovered in yeast [40] and mammalian [41] genomes and also identified in plant genomes of *Oryza sativa* [20,21], *Sorghum bicolor* [37], *Triticum aestivum* [42], *Gossypium* [43], *Brassica campestris* and *Brassica oleracea* [44]. In addition, gene conversion between duplicates or homoeologous chromosomes is frequent and long-lasting and has been demonstrated in genomes of the genus *Oryza* and the homoeologous chromosomes 11 and 12 of rice produced from the common tetraploidization event of grasses [21,36,37,45]. Although the preliminary inference of gene conversion between subgenomes produced from PRT in *A. hypogae**a* has been made [2], a comprehensive analysis of gene conversion for *Arachis* is lacking.

Mainly due to the biological and economic significance of *Arachis*, the genomes of five peanut species with different ploidies, including *A. duranensis* [3], *A. ipaensis* [6,46], *A. monticola* [5], *A. hypogaea* (Shitouqi) [2] and *A. hypogaea* (Tifrunner) [28], have been deciphered so far. Here, by performing a comparison analysis of these genomes, we aim to identify paralogous and orthologous gene sets associated with polyploidizations and species divergence, to assess the scale and patterns of conversion and to explore the factors that influence the occurrence of conversion, as well as its impact on genomic and functional evolution.

## 2. Materials and Methods

### 2.1. Genome Data

The wild diploid genomes of *A. duranensis*, *A. ipaensis* and the cultivated tetraploid Tifrunner were downloaded from Peanut Base (https://peanutbase.org/, accessed on 1 October 2020). The other one, the cultivated tetraploid Shitouqi genome, was obtained from PGR (http://peanutgr.fafu.edu.cn/, accessed on 1 October 2020), while the wild tetraploid *A. monticola* genome was downloaded from GIGA (http://gigadb.org/dataset/, accessed on 1 October 2020).

### 2.2. Detection of Duplicated Genes

To identify the duplicated genes produced by LCT and PRT and the orthologous genes related to the speciation of the considered *Arachis* genomes, BLASTP software [47] was first employed to search the potential homologous gene pairs, with the strict parameters of *e*-value < 1 × 10^−5^ and Score > 100. Then, the homologous gene information and locations on chromosomes were input into ColinearScan [14], to infer the colinear gene pairs and test the significance of the colinearity of chromosomal regions (blocks), while the key parameter, the maximum gap, was set to 50 intervening genes; the large gene families with 50 or more members were removed from the blocks. Lastly, we performed genomic homologous structure analyses through homologous dotplots to help to determine the paralogous and orthologous genes. This genome colinearity analysis approach was adopted in many previous angiosperm genomic comparisons [4,48,49].

### 2.3. Construction of Homologous Gene Quartets

Assessing the conversion between duplicate genes generated from LCT and PRT, we defined homologous gene quartets according to the gene colinearity information. If both genomes of any two *Arachis* species, A and B, retained one pair duplicate produced by LCT, a homologous gene quartet was formed by paralogous genes A1 and A2 from species A and their respective orthologous genes B1 and B2 from species B (Figure 1A). If there is no gene conversion between duplicated genes after species divergence, the sequence similarity between orthologous genes should be more similar than any pair of paralogous genes. However, if the duplicate genes are affected by conversion, we may find that the gene tree of the quartets exhibits a different structure compared to the expected topology (Figure 1B). To infer the conversion between duplicated genes located in different subgenomes of tetraploid peanut genomes, we constructed another type of quartet formed by one duplicated gene pair, Ama and Amb in *A. monticola* and their respective orthologous genes Ad in *A. duranensis* and Ai in *A. ipaensis*. Meanwhile, a similar approach was used to construct the quartets for the cultivated tetraploid *A. hypogaea* and its diploid ancestors of *A. duranensis* and *A. ipaensis* (Figure 1C,D).

### 2.4. Calculation of Ks and Ka

The synonymous nucleotide substitution rate (*Ks*) and nonsynonymous nucleotide substitution rate (*Ka*) between homologous gene pairs were estimated by using the Nei–Gojobori [50] approach, by implementing the program codeml in PAML [51]. ClustalW was employed to align multiple gene CDS and setting default parameters [52]. Due to the nucleotide substitutions frequently occurring at the same sites in a sequence, we used the Jukes–Cantor (JC) model to correct the *Ks* and *Ka* values, denoted by *Ps* and *Pa* [37,53].

### 2.5. Gene Conversion Inference 

ClustalW [54] was used to conduct multiple sequence alignment of amino acid sequences from each quartet. If the quartets had gaps in pair-wise alignment sequences accounting for >50% of the alignment length, or the amino acid identity of compared homologous genes was less than 40%, the quartet was removed. Those highly divergent quartets were removed to avoid the false inference of gene conversion resulting from problematic alignments.

Whole-gene conversion (WCV) inference: Since the divergence of orthologous gene pairs in a quartet occurred later than their respective paralogous genes, the expected similarity of orthologous gene pairs was higher than the paralogues in this quartet. However, the paralogous gene pairs may be affected by conversion and become more similar than their respective orthologous gene pairs. Here, we used two methods to infer the potential whole-gene conversion events, in which the similarity of homologous gene pairs was measured by *Ks* (defined as WCV-I). The bootstrap test was performed on the gene tree for each quartet to check the confidence level of the conversion events [20,37]. Additionally, we used the ratios of amino acid locus identity of sequences in each quartet to measure the similarity and examined of the topological tree changes to infer the potential whole-gene conversion events (defined as WCV-II). Compared with WCV-I, WCV-II has more stringent standards for inferring the conversion, because the divergence of these *Arachis* genomes occurred more recently. The similarity between orthologous sequences in different *Arachis* species is often very high due to their relatively close genetic relationship, as seen in previous studies of the conversion in hexaploid wheat and the genus *Oryza* [21,42].

Partial-gene conversion (PCV) inference: To detect possible gene conversion that affected only portions of a gene from paralogues, we employed a dynamic programming algorithm combined with phylogenetic analysis to search the DNA segments > 10 nucleotides in length affected by conversion, as in previous studies, to infer the partial-gene conversion of *Oryza* subspecies [21,22].

### 2.6. Statistical Analysis of the Correlation between Conversion and Physical Location

To check whether gene conversion is affected by the physical location of duplicated genes on chromosomes, we calculated the distance of duplicated genes relative to the chromosomal termini. Firstly, the duplicated genes on each chromosome arm were divided into 1 Mb bin run from the chromosome termini to the centromere and the number of duplicated genes in each bin was counted. Then, we divided the number of converted genes by the number of all duplicated genes to calculate the conversion rate in each bin. The fold increase in conversion rates was equal to the mean of the first selected bin divided by the mean of all other bins. Lastly, one million rounds of a permutation test were carried out by randomly swapping the box sums of the conversion rates and calculating the fold increase for each permutation, as previously reported [21,55].

### 2.7. Gene Ontology Analysis

InterProScan v5.0 [56] with default parameters was employed to identify the GO terms for each gene in *Arachis* genomes and the functional overview of duplicated genes is available. The online visualization tool WEGO (http://wego.genomics.org.cn/, accessed on 1 May 2021) [57] was used to compare and show GO annotation results of considered gene sets, while the functional distribution and changing trend of converted and nonconverted genes can be clearly displayed. The Pearson chi-squared test was used to test the significance of difference between the number of converted and nonconverted genes in the same biological function.

## 3. Results

### 3.1. Genomic Homology

Through the intra-genomic colinearity analysis, we inferred the gene colinearity within *Arachis* genomes. First, we identified the duplicated genes generated by recursive polyploidizations in each diploid and tetraploid peanut genome and found the *A. duranensis* with highly preserved intragenomic homology than other genomes (Appendix A). We identified 599 homologous blocks with four or more colinear genes, containing 5016 colinear gene pairs in *A. duranensis*. Using the same parameters, we found only 431 homologous blocks in *A. monticola* A, containing 2785 colinear gene pairs, which may be due to many chromosomal rearrangements occurred after it split from other species. Furthermore, we distinguished the blocks which were generated by LCT according to the median *Ks* of anchored gene pairs located in each block [26]. For example, within the dotplot of *A. ipaensis*, the homologous block between chromosomes 4 and 5, with a median *Ks* value of 0.86, was related to the LCT, indicates that this block was generated by LCT (Figure 2B and Appendix A). In this way, we obtained the duplicated gene sets which generated from LCT in different ploidy peanut genomes (Appendix A and Figure 2D). Ultimately, we found that the maximum number of duplicated genes was 2460 in *A. duranensis* and the minimum was 877 in *A. monticola* A.

To identify the orthologous genes between genomes, we performed inter-genomic comparisons of the considered genomes from five *Arachis* species (Appendix A). We found that there were 2170–2991 blocks preserved between two diploid peanut genomes, or between the subgenomes in tetraploid peanut genomes, involving 15,894–37,293 colinear gene pairs. Obviously, there is better genomic colinearity between genomes than within genomes. Furthermore, we identified the orthologous gene pairs between any two peanut genomes, according to the median *Ks* of anchored gene pairs located in blocks related to the divergence of genomes (or subgenomes) (Figure 2C and Appendix A). At last, we inferred that there were 5297–19,264 orthologous gene pairs between genomes or subgenomes, which were generated from the divergence of genomes (Appendix A).

### 3.2. Homologous Gene Quartets

Based on the above intra/inter-genomic homologous gene colinearity information, we constructed the quartets between the considered genomes. We identified 2200 quartets between *A. duranensis* and *A. ipaensis* and used these to infer the conversion events of duplicated genes generated from LCT that occurred after the divergence of these two diploid peanuts (Figure 1A). Then, we identified only 137 quartets between *A. monticola* A and *A. monticola* B, which we used to infer the conversions of duplicated genes generated from LCT which occurred after the formation of wild tetraploid peanut (Figure 1A). Additionally, we constructed 1315 quartets between *A. hypogaea* A (Shitouqi) and *A. hypogaea* B (Shitouqi) and 1954 quartets between *A. hypogaea* A (Tifrunner) and *A. hypogaea* B (Tifrunner), which are both related to the LCT events, and used them to infer the conversions occurred after the formation of the cultivated tetraploid peanut (Figure 1A). Fewer quartets were identified between two subgenomes of wild tetraploid peanut, possibly due to the extensive specific rearrangements occurred in its genome. Additionally, to infer the conversions between the duplicated genes generated from the PRT in tetraploid peanuts (Figure 1C), we identified the quartets between the subgenomes in *A. monticola*, *A. hypogaea* (Shitouqi) and *A. hypogaea* (Tifrunner) and two wild diploid peanut genomes, for which there were 2866, 10,784 and 13,306 quartets, respectively.

### 3.3. Gene Conversion between LCT-Related Duplicated Genes

By removing highly divergent quartets, we obtained the reliable quartets for further inferring the gene conversion (Table 1). After filtering, we successfully identified the 1871 quartets between *A. duranensis* and *A. ipaensis* and the quartets among the subgenomes in three tetraploid peanut genomes of *A. monticola*, Shitouqi and Tifrunner, amounting to 99, 1314 and 1935, respectively (Table 1). Using these quartets, we inferred the whole-gene (WCV-I and WCV-II) and partial-gene conversion (PCV) events between duplicates through the comparison of the gene tree topology changes, based on the similarity of the synonymous nucleotide substitution rate and the amino acid identity rate (see the Methods for details).

In *A. duranensis*, we found that 220 (11.8%) of the paralogues were converted after this species’ divergence from *A. ipaensis*, among which 4 (0.2%) of the paralogues were affected by WCV and 216 (11.6%) of the paralogues were affected by PCV (Table 1 and Figure 3A). In *A. ipaensis*, we found the 242 (13.0%) of the paralogues were converted after this species’ divergence from *A. duranensis*, among which only 2 (0.1%) of the paralogues were affected by WCV and 241 (12.9%) of the paralogues were affected by PCV (Table 1 and Figure 3A). Considering that the conversions could be affected by the PRT, we inferred the conversion between the duplicated genes related to LCT in the subgenomes of *A. monticola* (Table 1 and Figure 3B). We found that 13 (13.1%) of the paralogues were converted in *A. monticola* A and 14 (13.1%) of the paralogues were converted in *A. monticola* B, which are similar to the conversion rates in two diploid peanuts, but show signs of being slightly higher.

In addition, considering that the conversions could be affected by PRT and artificial domestication, we further independently inferred the conversion of the duplicated genes related to LCT in the subgenomes of Shitouqi and Tifrunner. We found that slightly fewer duplicated genes were affected by gene conversion in Shitouqi, with 126 (9.6%) of the paralogues having been converted in subgenome A and 115 (8.8%) of the paralogues having been converted in subgenome B (Table 1 and Figure 3C). Similar to Shitouqi, we found that fewer duplicated genes were affected by conversion in Tifrunner, with 132 (6.8%) of the paralogues having been converted in subgenome A and 139 pairs (7.1%) of the paralogues having been converted in subgenome B (Table 1 and Figure 3D). These conversion rates for paralogues in different ploidies of the peanut genome suggest that habitats and genetic bases both have a certain influence on the occurrence of conversion.

### 3.4. Gene Conversion between PRT-Related Duplicated Genes

To infer the conversions between PRT-related duplicated genes, we also removed the highly divergent quartets. After filtering, we obtained the quartets between the subgenomes of *A. monticola*, Shitouqi and Tifrunner, and two diploid peanut genomes, amounting to 2625, 9972 and 12,657, respectively (Appendix A). Then, using these quartets, we explored the gene conversion between subgenomes of tetraploid peanut. In *A. monticola*, we inferred that 433 (16.5%) of duplicated gene pairs related to PRT were affected by gene conversion. Of these, the conversion patterns of 263 (10.0%) of the paralogues were inferred to be WCV-I and 354 (13.5%) of the paralogues were inferred to be WCV-II, whereas there were fewer paralogues affected by PCV, with only 5 (0.2%) pairs. Meanwhile, we found that the 181 (54%) of the converted genes in *A. monticola* located in subgenome A were used as donors; the other converted genes used as donors were located in subgenome B.

Similarly, we also inferred the conversion between duplicated genes related to PRT in two cultivated tetraploid peanuts. In Shitouqi, we detected 1122 (11.3%) of the PRT-related duplicated genes affected by gene conversion. Of these, 607 (6.1%) of the duplicates were inferred to be WCV-I and 1006 (10.1%) of the duplicates were inferred to be WCV-II, with only 9 (0.1%) of the duplicates having been affected by PCV (Figure 4A and Appendix A). In Tifrunner, we detected 1706 (13.5%) of the PRT-related duplicated genes affected by gene conversion. Of these, 1115 (8.8%) of the duplicates were inferred to be WCV-I and 1495 (11.8%) of the duplicates were inferred to be WCV-II, with only 29 (0.2%) of the duplicates having been affected by PCV (Figure 4B and Appendix A). Comparing the results of the above inferences, we found that the conversion between duplicates located in different subgenomes of wild tetraploid peanut occurs with a higher frequency than in cultivated tetraploid peanut, while the dominant source of donors is also different in two tetraploid peanuts.

Furthermore, we compared the distribution of donors in the subgenomes of Shitouqi and Tifrunner and found that 112 (41%) of the converted genes located in subgenome A and 158 (59%) of the converted genes located in subgenome B were inferred to be donors. Interestingly, 84.7% (222) of the genes were taken as donors in the two genomes. This suggests that the donor genes in the converted duplicated genes are often taken as donors in different genomes. For example, part of an orthologous gene pair, *Sha03g5270* of Shitouqi and *Tha03g4128* of Tifrunner were both found to be donors in two tetraploid peanut genomes (Figure 4C,D).

### 3.5. Conversion and Evolution

Conversion homogenizes paralogous gene sequences, which makes those paralogues affected by conversion appear younger than expected based on sequence divergence with one another [20,21,37,58]. Here, we also found that the average *Pn*  =  0.181 and *Ps*  =  0.534 of converted paralogues were significantly smaller than the average *Pn*  =  0.199 and *Ps*  =  0.559 of nonconverted converted paralogues in *A. duranensis* (*p*-value  =  6.32 × 10^−3^, *p*-value  =  6.13 × 10^−4^, *t*-test) (Appendix A). However, this comparison could not determine whether converted genes evolved slowly based on the paralogues themselves, since the pairwise distances between paralogues was distorted by conversion. Thereby, we further compared the *Pn* and *Ps* of converted and nonconverted orthologues between the considered genomes and found the distance of orthologues affected by conversion to be significantly larger than that of those orthologues not affected by conversion (Table 2 and Appendix A). For example, the average *Pn* = 0.055 and *Ps* = 0.109 of converted paralogues between *A. duranensis* and *A. ipaensis* were significantly larger than the average *Pn* = 0.023 and *Ps* = 0.064 of paralogues not affected by conversion (*p*-value = 9.48 × 10^−14^, *p*-value = 7.48 × 10^−7^, *t*-test). These results suggest that the converted paralogues have evolved faster than the nonaffected ones, also indicating the conversion contributes to the divergence of genus *Arachis*.

To determine whether the gene conversion was affected by evolutionary selection pressure, we employed the *Pn*/*Ps* ratios of paralogues and orthologues to reflect the selection pressure during their evolution (Appendix A). The average *Pn*/*Ps* ratio of converted paralogues in *A. duranensis* was 0.34, similar to the average *Pn*/*Ps* ratio of nonconverted paralogues, 0.36. This comparison seems to suggest that the conversion does not result in obvious changes in selection pressure of the paralogues in *A. duranensis*. Similarly, to check the corrections of conversion and evolutionary rates, we further used the *Pn*/*Ps* ratios of orthologues to find the actual selection pressure difference. We found that the average *Pn*/*Ps* ratio of converted orthologues between *A. duranensis* and *A. ipaensis* was 0.505, significantly larger than the average *Pn*/*Ps* ratio of nonconverted orthologues, which was 0.359 (3.68 × 10^−9^) (Table 2). This difference in *Pn*/*Ps* ratio also exists in comparisons between other genomes (Table 2). These results suggest that conversion reduces the negative selection pressure on genes, making them prone to the “free” mode of evolution.

### 3.6. Conversion and Physical Position

By calculating the rate of gene conversion occurring on different chromosomes, we found no significant difference in the rate of gene conversion between different chromosomes (Appendix A). For example, the average gene conversion rate of 10 chromosomes in the *A. duranensis* genome was 11.8%, while the conversion rate of each chromosome was distributed in the smaller range of 9.7~14.1%, with no significant difference (*p*-value = 0.998). Furthermore, we found that the distribution of converted genes was unbalanced in the different regions of each chromosome (Appendix A), as the converted genes tended to be located in near the end of the chromosome (Figure 3). For instance, approximately 30% of all converted genes generated from the PRT event were located within 5% of the end of the chromosomes. However, we did not find a high rate of gene conversion near the end of the chromosome (Appendix A). The average rate gene conversion of *A. duranensis* genome was 11.8%, similar to the rate of conversions within the 5 Mb region near the chromosomal telomeres, which was 13.1%.

### 3.7. Chromosome Rearrangements and Conversion

Chromosome rearrangement events possibly disrupt genomic collinearity and the degree of chromosome rearrangement can be reflected by the number of blocks in the genome. To explore potential associations between rearrangements and conversion, we investigated the relationship between the conversion rate and the numbers of blocks related to the LCT and PRT event on each chromosome from eight *Arachis* genomes, respectively (Appendix A). After a thorough comparison, unfortunately, we found no valuable correlation; even if there was a hint of correlation, the tendency was inconsistent across genomes. For example, we found that the gene conversion rate was weakly negatively correlated with the number of blocks in *A. duranensis* (R^2^ = 0.0681), whereas a weakly positively correlation was exhibited in *A. ipaensis* (R^2^ = 0.0052). Furthermore, when investigating the relationship between the length (colinear gene pairs) of the colinearity region and the gene conversion rate, it was found that the conversion of longer regions showed a higher rate than the shorter regions (Figure 5), while, in the homologous chromosomal regions with more than 50 gene pairs, the gene conversion rate in *A. duranensis* was 13.1%, smaller than the conversion rate of 4.3% in those regions of with less than 10 gene pairs (Appendix A). Although no correlation between rearrangement and conversion was found, we still revealed that the well-preserved ancestral homology can facilitate gene conversion.

### 3.8. Gene Function Analysis

The probability of a gene being converted may be associated with its function; thus, we performed a gene ontology analysis to identify the GO terms for duplicated genes in the studied peanut genomes. Firstly, we identified the GO terms of 235, 207, 240 and 262 LCT-related converted genes in *A. hypogaea* A (Shitouqi), *A. hypogaea* B (Shitouqi), *A. hypogaea* A (Tifrunner) and *A. hypogaea* B (Tifrunner), respectively. Comparing the proportion of converted and duplicated genes for each function, we found that some genes with specific functions were more likely to be converted, whereas there were some functional genes that were biased toward escape from conversion (Figure 6). We found that the genes involved in those functions associated with large numbers of genes were biased towards gene conversion (Appendix A). For example, regarding the catalytic activity-related genes in *A. hypogaea* A (Tifrunner), the converted genes accounted for 23.1% of all converted genes, a significantly higher level than that of the duplicated genes related to this function, which only accounted for 15.1% of all duplicated genes in the whole genome (*p*-value < 0.001). This implies that catalytic activity-related genes tended to be affected by conversion. In contrast, some genes associated with functions (regulation of metabolic process) encoded by few genes might have avoided conversion (Figure 6).

Furthermore, we checked the domains involved in the converted genes in all the studied peanut genomes. Duplicated genes that had experienced gene conversion in diploid peanut were enriched in helix–loop–helix DNA-binding domain (*p*-value = 1.23 × 10^−6^), WD (*p*-value = 6.65 × 10^−5^) and ring finger domain (*p*-value = 4.13 × 10^−3^) (Figure 7A and Appendix A). After the formation of tetraploid peanuts, the domains involved in the converted genes of Shitouqi were enriched in the triose-phosphate Transporter family (*p*-value = 2.20 × 10^−16^), the helix–loop–helix DNA-binding domain (*p*-value = 4.55 × 10^−4^) and protein phosphatase 2C (*p*-value = 3.09 × 10^−4^) (Figure 7B and Appendix A). The domains involved in the converted genes of Tifrunner were enriched in the helix–loop–helix DNA-binding domain (*p*-value = 3.93 × 10^−4^), the RNA recognition motif (*p*-value = 7.62 × 10^−2^) and short chain dehydrogenase (*p*-value = 8.74 × 10^−6^) (Figure 7C and Appendix A). Among all the genomes of different ploidy peanut, the converted genes involved domains which were enriched in the helix–loop–helix DNA-binding domain (*p*-value = 2.20 × 10^−16^), Ring finger domain (*p*-value = 2.61 × 10^−6^) and the RNA recognition motif (*p*-value = 6.01 × 10^−3^) (Figure 7D and Appendix A). These results suggest that the identified converted genes with specific domains may be associated with important traits of peanut growth and development.

## 4. Discussion

### 4.1. Long-Lasting Extensive Conversions Affected the Evolution of Duplicated Genes in Peanut Genomes

Duplicated genes generated from recursive ancient polyploidizations, which played an important role during the diversification of green plants, have been reported in many previous studies [10,11,12,59,60,61,62,63,64]. Here, we inferred the conversions between duplicated genes produced by LCT and PRT events and offered new insights into the evolutionary process of duplicated genes in peanut genomes. First, duplicated genes have been produced for a long time and they still interact with each other as a high frequency under the action of illegitimate genetic recombination, as demonstrated by the conversion between LCT-produced duplicates here identified and by the findings of previous studies on sorghum and rice [37]. Second, the conversion affects the DNA sequence to varying degrees, either at the level of the entire gene, or at that of only a few nucleotide sites. Third, conversion is an on-going long-lasting event which affected the evolution of duplicated genes, here revealed by the conversion events that occurred between the duplicates produced by the recent duplication event (PRT) of tetraploid peanut genomes.

### 4.2. Conversion Contributes to the Divergence of Genus Arachis Genomes

Conversions cause the sequence of duplicated gene pairs to become more similar than expected and it seems that the conversion causes the duplicates to be more conserved. However, we found that conversion accelerates the evolutionary rate of duplicate genes and contributes to the divergence of genus *Arachis* genomes. The main reason for this apparent result is that conversion distorted the genetic distance between duplicated gene pairs; this has also been demonstrated in previously studies [21,65,66]. Here, we emphasize that a closer understanding of the effect of conversion on the rate of nucleotide evolution should be obtained by comparing orthologous gene pairs between genomes. In addition, gene conversion as an accelerating force of nucleotide variation may lead to the transfer of a new mutation from one gene in duplicates to another copy, accelerating the divergence of peanut genomes.

### 4.3. Donor Genes Are Preferred as Donors

A gene conversion event involves copying one gene sequence from a donor locus to a receptor locus [67]. As a consequence of the conversion, the “acceptor” sequence is replaced, wholly or partly, by a sequence that is copied from the “donor”, whereas the sequence of the donor remains unaltered. This gene conversion pattern has also long been identified in mammalian cells, with the human hemoglobin genes of HBG1 and HBG2 being the first characterized examples [41]. Comparative analyses of the characteristics of donor and acceptor genes in conversion events are helpful for elucidating the mechanism of conversion. We found that independent conversion events that have survived (so far) in different peanut genomes often used the same genes as donors. It seems improbable to attribute this to selection, as a gene from the ancestor of a diploid peanut wild species was inherited from two different varieties of tetraploid peanuts and was mostly consistently expressed as a donor. This indicates that a gene affected by gene conversion in one species as a donor is usually preferred as a donor in another species if it is also affected by conversion. A more plausible explanation is that one gene copy has a “privileged” nature over the other. This could be genetic or epigenetic. If one gene or its neighboring region possesses mutations or epigenetic changes, the other gene might be more likely to act as a donor, helping to reinstate intactness.

### 4.4. Conversion and Genomic Rearrangements

Duplicated genes distributed near the ends of chromosomes tend to undergo conversion, which has been reported in rice, sorghum, genus *Oryza* genomes and hexaploid wheat [20,21,37,42,45]. However, in peanut genomes, we did not find the duplicated genes near the end of the chromosome to be more preferentially converted. If gene conversion is based on interactions between similar DNA sequences, this finding seems unreasonable for the following reasons. First of all, most duplicates are distributed in regions near the end of chromosomes or far from the centromere [26,68] and the DNA sequence should have higher similarity, which can provide suitable basic conditions for the occurrence of conversion. In contrast, the abundance of repeat elements near the centromere often increases the frequency of DNA rearrangement and nucleotide variation, which ultimately leads to a reduction in the sequence similarity between homoeologous chromosomes related to the WGDs. Repetition elements are enriched in the pericentromeric regions, which has been demonstrated in many angiosperm genomes, such as rice, sorghum, cotton, soybean and peanuts [2,4,6,25,27,43,69,70]. Through careful examination, we found that some duplicates distributed near the terminal regions of chromosomes still showed a preference for conversion, involving the chromosomes 2, 4, 5 and 6 in tetraploid peanut (Tifrunner) genomes, which maintained a good ancestral genomic structure (Figure 4B and Appendix A). Additionally, we also found that the length (colinear gene pairs) of the blocks may be positively correlated with the conversion rate, that is, the well-preserved homoeologous regions showed a higher conversion rate. This suggests that the duplicates located near the chromosomal terminal regions were not preferentially converted in the peanut genome, which may be caused by extensive genomic rearrangements after LCT and PRT events [2,26]. Genomic rearrangements can change the structure of ancestral chromosomes and the gene collinearity between homologous chromosomes produced by WGDs is often destroyed [71,72,73]. There were more genome rearrangements in peanut genomes than in rice and sorghum genomes relative to the ancestral genomes of their respective families (Legume and Poaceae) [2,26,72]. Perhaps, as a result of polyploidizations, the terminal region of the ancestral chromosome may no longer be near the telomeres in the peanut genome. In the future, we can further explore whether the regions that maintain good genomic collinearity and are preferentially affected by conversion are the regions near the end of ancestral chromosomes.

### 4.5. Conversion and Function

Gene conversion causes duplicated gene pairs to be very similar or even identical in sequence and the presence of duplicate copies may neutralize meaningful mutations and provide opportunities for functional innovation [74]. The evolution of functional genes that are members of large families may often be accompanied by strong purifying selection, as proposed by previous studies [75,76,77,78,79,80,81]. We confirmed that the functions associated with multigene families may be biased toward the occurrence of gene conversion. These results are also consistent with previous studies which proposed that the most multigene families were thought to have coevolved with related homologous genes through gene conversion [82]. In addition, gene conversion is emerging as a driver of innovation amongst meiotic drive genes, which likely contributed to the expansion and birth of meiotic driver genes [83,84]. This may be especially true when important components of drive systems consist of segments of DNA that can be copied multiple times within a genome. Here, we found that the effect of conversion on the functional genes in diploid ancestors and tetraploid peanut was inconsistent, even in different tetraploid cultivars peanuts which were inconsistent in certain functions; this may be due to the fact that geographical distribution and artificial domestication may have caused the two different varieties of peanuts to evolve in different directions.

## 5. Conclusions

Duplicated genes in *Arachis* genomes generated from recursive polyploidizations experienced long-lasting effects from gene conversion. By performing comparative genomics and phylogenetic analyses, we identified the scale and patterns of conversion between duplicates produced by LCT and PRT events during the diversification of the genus *Arachis*. Gene conversion maintained the similarity of duplicate sequences, provided opportunities for further gene conversion and accelerated the evolutionary rate of *Arachis* genomes. Chromosome rearrangements after polyploidization are associated with gene conversion events, while the well-preserved homoeologous chromosome regions may facilitate the conversion of duplicate genes. The genes involved in the functions associated with multigene families may be preferentially converted. We identified specific domains which were involved in converted genes, implying that conversions are associated with important traits of peanut growth and development. This present effort will contribute to understanding the evolution of duplicated genes affected by gene conversion in *Arachis* genomes.

## Figures and Tables

**Figure 1 genes-12-01944-f001:**
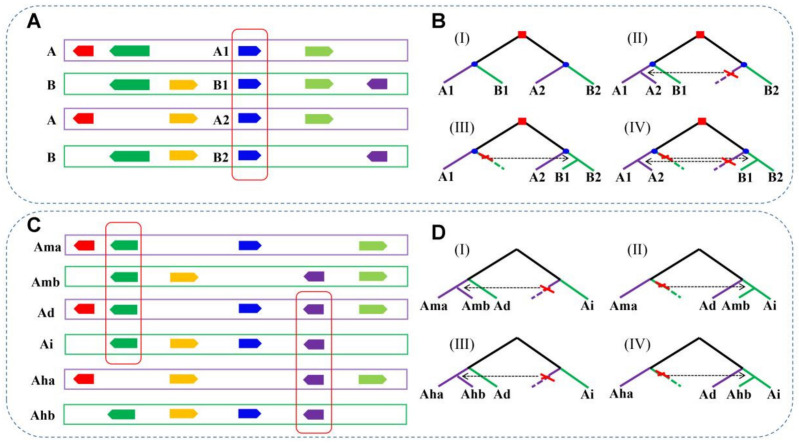
Homologous gene quartets and inference of conversion through phylogenetic analyses. (**A**) Colinear chromosomal segments from two genomes (**A**,**B**), represented by rectangles of different colors. Arrows show genes and homologous genes are coded by the same color. Homologous gene quartet formed by paralogous genes A1 and A2 in A and their respective orthologous genes B1 and B2 in B. (**B**) Squares indicate the WGD event in their common ancestral genome and circles symbolize species divergence. The expected phylogenetic relationship and potential conversion event of the homologous quartets: (I) the expected phylogenetic relationship of the homologous genes in quartet if no conversion occurs; (II) A2 (acceptor) is converted by A1 (donor); (III) B1 is converted by B2; (IV) both of the above conversions occurred. (**C**) Homoeologous chromosomal segments from *A. duranensis*, *A. ipaensis*, *A. monticola* and *A. hypogaea*. (**D**) The phylogenetic relationship of conversions between PRT-related duplicated genes.

**Figure 2 genes-12-01944-f002:**
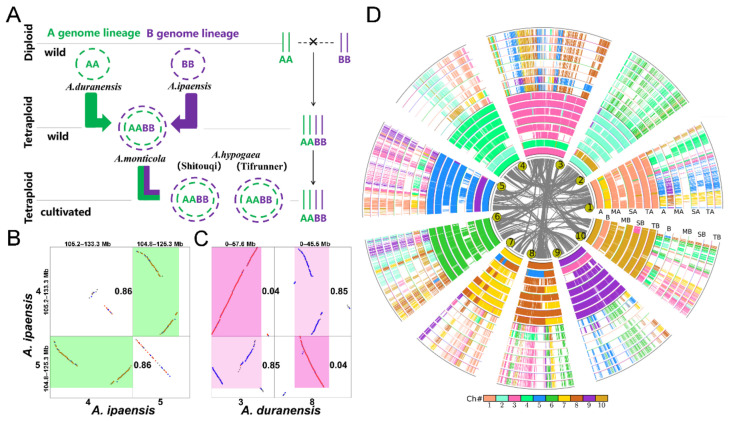
The phylogeny of the studied peanut genomes and inference of duplicated and orthologous gene pairs within and between genomes. (**A**) The phylogeny of the studied peanut genomes. (**B**) The genomic homologous dotplot within the *A. ipaensis* genome. Highlighted boxes indicate the paralogous blocks generated from the LCT event. The median *Ks* values of anchored gene pairs located in blocks are placed next to the highlighted boxes. (**C**) The genomic homologous dotplot between the genomes of *A. ipaensis* and *A. duranensis*. Highlighted boxes indicate the orthologous blocks produced by the divergence of *A. ipaensis* from *A. duranensis*. The median *Ks* values of anchored gene pairs located in blocks is placed next to the highlighted boxes. (**D**) Alignment of the peanut and relative genomes with *A. duranensis* as reference. The innermost circle represents the 10 chromosomes of the *Ad* genome and the gray lines linked paralogous genes generated from LCT. Genomic paralogy, orthology and outparalogy information within and among eight (sub) genomes, with the name abbreviations of *A. duranensis* (A), *A. ipaensis* (B), *A. monticola* A (MA), *A. monticola* B (MB), *A. hypogaea* A (Shitouqi) (SA), *A. hypogaea* B (Shitouqi) (SB), *A. hypogaea* A (Tifrunner) (TA) and *A. hypogaea* B (Tifrunner) (TB), displayed in 16 circles. The short line forming the innermost *A. duranensis* chromosome circle represents predicted genes, which have one set of paralogous regions, forming another circle. Each of the two sets of *A. duranensis* paralogous chromosomal regions has one orthologous region in other *Arachis* genomes.

**Figure 3 genes-12-01944-f003:**
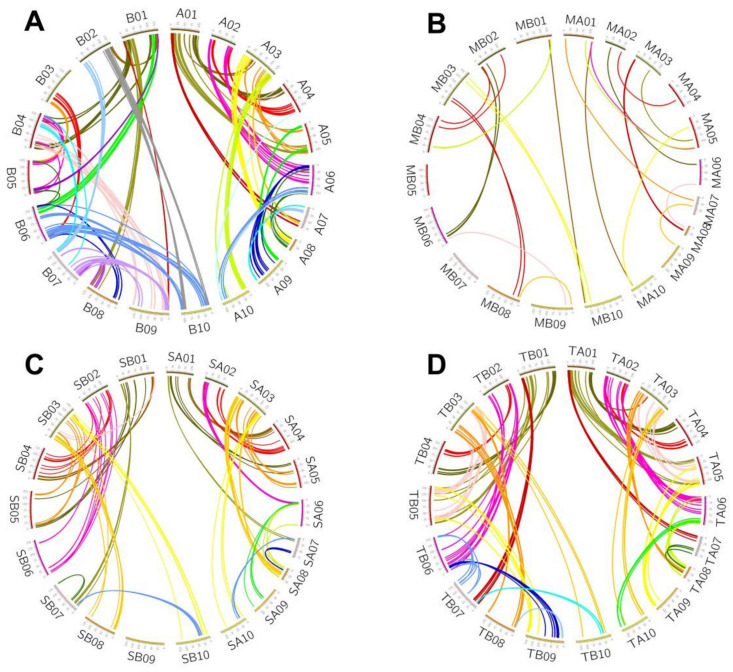
Gene conversion in peanut genomes. In each panel, the outer circle shows the chromosomes in the considered peanut genome. Converted duplicated gene pairs are connected with curvy lines. (**A**) Gene conversion in *A. duranensis* and *A. ipaensis*. (**B**) Gene conversion in *A. monticola*. (**C**) Gene conversion in Shitouqi. (**D**) Gene conversion in Tifrunner.

**Figure 4 genes-12-01944-f004:**
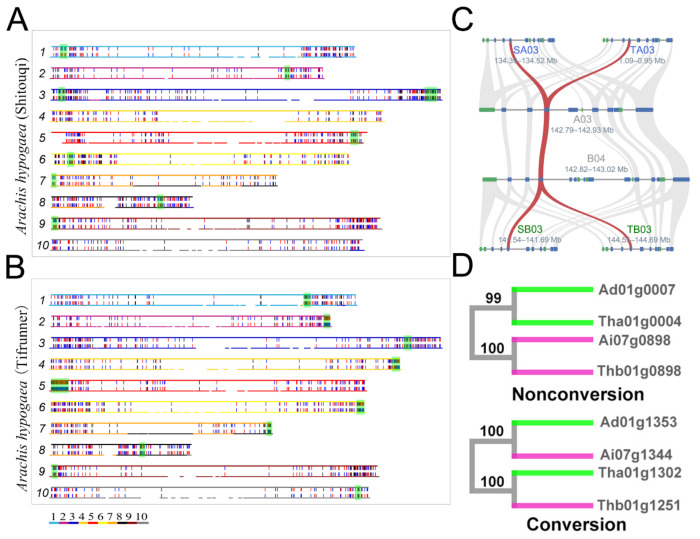
Gene conversions between PRT-related duplicated genes in two cultivated tetraploid peanuts genomes. (**A**) Twenty chromosomes of Shitouqi are divided into ten groups according to the corresponding relationship of the subgenomes in Shitouqi. Converted duplicated genes related to PRT are marked out with a short line according to their location on subgenome A and B, while the red lines indicate genes from converted duplicates as donors and the blue lines indicate acceptor genes; the highlighted region indicates that the number of converted genes is no less than 25% of duplicates in the region. (**B**) Gene conversions between PRT-related duplicated genes in Tifrunner genome. (**C**) Colinearity and conversion at the end of chromosome 3 in Shitouqi and Tifrunner; rectangles represent annotated genes with orientation on the same strand (blue) or reverse strand (green); the grey lines connect syntenic gene pairs and red lines connect conversion genes. (**D**) The top is the tree of nonconverted genes and the bottom is the converted gene tree.

**Figure 5 genes-12-01944-f005:**
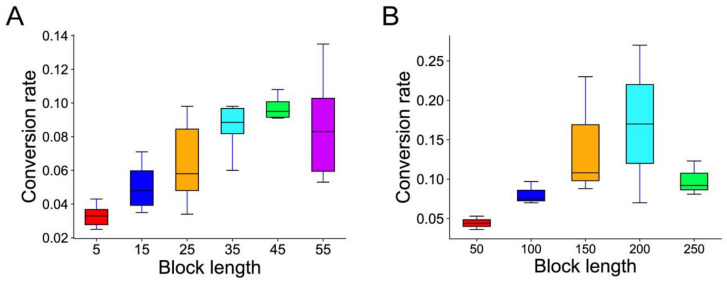
The association between the block length (colinear gene pairs) and the gene conversion rate. (**A**) The association between LCT-related block length and conversion rate among all eight peanut genomes. (**B**) The association between PRT-related block length and gene conversion among three tetraploid peanut genomes.

**Figure 6 genes-12-01944-f006:**
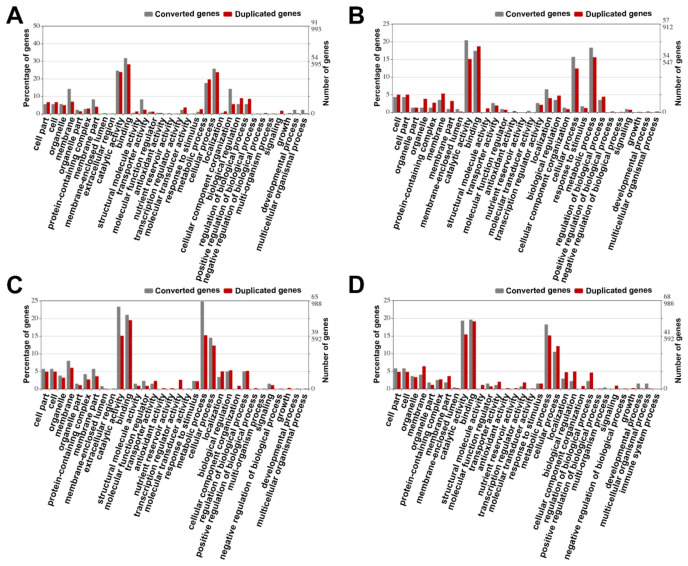
Histogram of gene ontology (GO) statistics for converted and all duplicated genes. (**A**) GO statistics for converted genes and duplicated genes in *A. hypogaea* A (Shitouqi). *x*-axis shows user-selected GO terms; *y*-axis shows the percentages of genes (number of a particular gene divided by total gene number). (**B**) GO statistics for converted genes and duplicated genes in *A. hypogaea* B (Shitouqi). (**C**) GO statistics for converted genes and duplicated genes in *A. hypogaea* A (Tifrunner). (**D**) GO statistics for converted genes and duplicated genes in *A. hypogaea* B (Tifrunner).

**Figure 7 genes-12-01944-f007:**
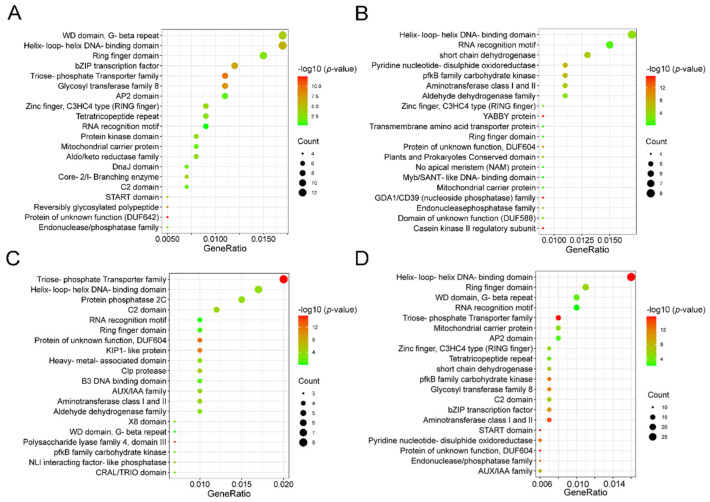
Bubble diagram of domain enrichment for converted genes. (**A**) Enrichment of duplicated gene domains in gene conversion in diploid peanut. The abscissa represents the total percentage of genes containing this domain, the ordinate represents the various domains; the size of the circle represents the number and the color represents the *e*-value. (**B**) Enrichment of duplicated gene domains in gene conversion in Shitouqi. (**C**) Enrichment of duplicated gene domains in gene conversion in Tifrunner. (**D**) Enrichment of duplicated gene domains in gene conversion in all the different ploidies of peanut.

**Table 1 genes-12-01944-t001:** Converted paralogues in peanut genomes.

Species	Quartet Patterns	Paralogues in Quartets	WCV-I ^a^	WCV-II ^b^	PCV ^c^	Total	Conversion Rate (%)
*A. duranensis*	A_1_-B_1_-A_2_-B_2_	1871	-	4	216	220	11.8%
*A. ipaensis*			-	2	241	242	13.0%
*A. monticola* A	Ama_1_-Amb_1_-Ama_2_-Amb_2_	99	-	2	11	13	13.1%
*A. monticola* B			-	-	14	14	14.1%
*A. hypogaea* A (Shitouqi)	Aha_1_-Ahb_1_-Aha_2_-Ahb_2_	1314	3	3	121	126	9.6%
*A. hypogaea* B (Shitouqi)			3	2	112	115	8.8%
*A. hypogaea* A (Tifrunner)	Aha_1_-Ahb_1_-Aha_2_-Ahb_2_	953	2	2	129	132	6.8%
*A. hypogaea* B (Tifrunner)			3	3	134	139	7.1%

Note: WCV-I ^a^, the similarity of homologous gene pairs measured by *Ks*; WCV-II ^b^, the ratios of amino acid locus identity of sequences in each quartet to measure the similarity and examination of the topological tree changes; PCV ^c^, a dynamic programming algorithm combined with phylogenetic analysis.

**Table 2 genes-12-01944-t002:** Nucleotide substitution rates of quartets in peanut genomes.

Orthologues		Converted Genes	Nonconverted Genes	*p*-Value (*t*-Test)
*A. duranensis–A. ipaensis*	*Pn*	0.055	0.023	9.48 × 10^−14^
	*Ps*	0.109	0.064	7.05 × 10^−7^
	*Pn*/*Ps*	0.505	0.359	3.68 × 10^−9^
*A. monticola* A–*A. monticola* B	*Pn*	0.114	0.0380	1.40 × 10^−3^
	*Ps*	0.220	0.0846	3.60 × 10^−3^
	*Pn*/*Ps*	0.523	0.449	1.30 × 10^−3^
*A. hypogaea* A (Shitouqi)–*A. hypogaea* B (Shitouqi)	*Pn*	0.0620	0.0483	2.10 × 10^−3^
	*Ps*	0.116	0.0900	9.00 × 10^−3^
	*Pn*/*Ps*	0.534	0.537	4.98 × 10^−5^
*A. hypogaea* A (Tifrunner)–*A. hypogaea* B (Tifrunner)	*Pn*	0.0704	0.0340	4.01 × 10^−5^
	*Ps*	0.133	0.0753	4.10 × 10^−5^
	*Pn*/*Ps*	0.529	0.452	9.33 × 10^−7^

## Data Availability

The datasets supporting the conclusions of this article are included within the article.

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
