# Peer review of "Illegitimate Recombination between Duplicated Genes Generated from Recursive Polyploidizations Accelerated the Divergence of the Genus Arachis"

_genes, 2021, doi:10.3390/genes12121944_

Round 1

Reviewer 1 Report

The manuscript, Illegitimate recombination between duplicated genes generated from recursive polyploidizations accelerated the divergence of genus Arachis; by Shen et al. is a nice attempt to attempt to understand how polyploidizations accelerated the divergence of genus Arachis.  They have compared five genomes with different ploidy levels to find that gene conversions maintain the similarity of duplicated genes. They also discussed how genes with different functions preferentially get converted. Overall, the analyses and results are presented well. The results section could be improved by identifying the main conclusion of each section and clearly mentioning that.

However, I believe the discussion section needs to be improved significantly. The purpose of the discussion is to relate the finding of your study to what’s already known in the field. This is very minimal in the discussion. Also, I am not sure what the purpose of the first paragraph of the discussion is. Should consider modifying it or removing it.

Other specific comments:

Why do genes with certain functions get preferentially converted? Could this be a positional effect rather than a functional effect?

Species names should be in italics throughout the manuscript.

Reviewer 2 Report

In the research article “Illegitimate recombination between duplicated genes generated from recursive polyploidizations accelerated the divergence of genus Arachis”, the authors have performed synteny-based comparisons of two diploid and three tetraploid peanut genomes. Then they have identified the duplicated genes generated from LCT and peanut recent PRT within genomes. Also, the authors detected conversion between duplicates, and that those are unevenly distributed across the chromosomes. Finally, the authors identified the biological functions that contain a higher number of genes and are converted preferentially. The authors did undertake an interesting approach, and the methodology was relevant. However, the authors should pay a lot of attention to improving the manuscript on the writing style especially improving the language and the grammar. Many sentences are not written properly, and the writing makes it hard to understand the sentences correctly. I would recommend editing the manuscript by a native English speaker.